# Clinical Characteristics, Serum Biochemical Changes, and Expression Profile of Serum Cfa-miRNAs in Dogs Confirmed to Have Congenital Portosystemic Shunts Accompanied by Liver Pathologies

**DOI:** 10.3390/vetsci7020035

**Published:** 2020-03-25

**Authors:** Ahmed M. El-Sebaey, Pavel N. Abramov, Fatma M. Abdelhamid

**Affiliations:** 1Department of Clinical Pathology, Faculty of Veterinary Medicine, Mansoura University, Mansoura 35516, Egypt; fatmamostafa980@yahoo.com; 2Department of Diagnostics of Diseases, Therapy, Obstetrics and Animal Reproduction, Moscow State Academy of Veterinary Medicine and Biotechnology – MVA named K. I. Skryabin, 109472 Moscow, Russia; abramov_p@inbox.ru

**Keywords:** canine, computed tomography angiography, congenital portosystemic shunts, liver pathologies, serum biochemical changes, serum cfa-miRNAs

## Abstract

Computed tomography angiography (CTA) and biochemical parameters cannot specify liver pathologies in dogs with congenital portosystemic shunts (CPSS) that are easily determined by invasive histopathology. This study aims to assess the possibility of using circulating serum canine familiaris (cfa) microRNAs (miRNAs) as novel non-invasive serum-based fingerprints for liver injuries associated with various morphologies of extrahepatic and intrahepatic portosystemic shunts (EHPSS and IHPSS). Data were obtained from 12 healthy dogs and 84 dogs confirmed to have EHPSS (splenocaval, splenophrenic, splenoazygos, right gastrocaval (RGC), right gastrocaval with caudal loop (RGC–CL)) and IHPSS (right divisional and left divisional) using CTA. Hepatic pathologies were determined by histopathology. Serum expression of miRNAs was assessed by real-time polymerase chain reaction. Based on the nature of liver injuries in each shunt type, cfa-miR-122 was significantly upregulated in all CPSS groups. Meanwhile, serums cfa-miR-34a and 21 were not significantly expressed in splenophrenic or splenoazygos groups, but they were extensively upregulated in splenocaval, RGC, RGC–CL groups and less frequently in right or left divisional groups. Also, serum cfa-miR126 was significantly upregulated in both IHPSS groups but less significantly expressed in RGC, RGC–CL, and splenocaval groups. Overall, estimated cfa-miRNAs could serve as novel biomarkers to mirror the histopathological and molecular events within the liver in each shunt type.

## 1. Introduction

Congenital portosystemic shunts (CPSS) are considering the most frequent vascular abnormality of the hepatobiliary system in dogs (*Canis lupus familiaris*). CPSS conjoins the portal vein with the caudal vena cava (CVC) or azygos vein, allowing blood from the portal circulation (incorporated with toxins, hepatotrophic substances, and nutrients) to bypass the hepatic parenchyma and flow directly into the circulation system [1]. Subsequently, inducing hepatic dysfunction [2], as well as circulatory persistence of neurotoxic substances typically metabolized by the liver, such as ammonia, give rise to a wide variety of clinical signs and serum biochemical and histological abnormalities [3,4]. Thereupon, the surgical attenuation of the shunting vessel is the gold standard to redirect portal blood into hepatic tissue to promote normalization of hepatic structure and function [5].

Depending upon ultrasonography and computed tomography angiography (CTA), CPSS are commonly confirmed either as extrahepatic or intrahepatic portosystemic shunts (EHPSS or IHPSS) [6]. In spite of this, diagnostic imaging techniques together with standard liver chemistry tests do not truly specify the nature of ongoing liver pathologies and fail to reflect the ability of hepatic tissue to accommodate with sudden portal blood flow after shunt attenuation that could be achieved by microscopic configuration of hepatic tissue [7,8]. Unfortunately, liver biopsies usually obtained during shunt attenuation cannot hence be used for clinical decision-making before surgical intervention according to the severity of histological changes [9]. Also, the results of histologic examination may not parallel to the degree of liver damage due to sampling variation; the results also cannot be used routinely as an invasive tool because they carry some risk [5]. Thus, the development of a reliable serum-based fingerprint of ongoing liver injury has long been desired for either clinical decision-making or case monitoring over time.

Mature microRNAs (miRNAs) are a class of short (∼18–25 nucleotides in length), noncoding RNAs molecules that act as crucial regulators of post-transcriptional gene expression in diverse physiological and pathological pathways related to hepatocytes proliferation, metabolism, oxidative stress, angiogenesis, steatosis, apoptosis, necrosis, and fibrosis [10]. Recent studies revealed that the serum and tissue concentrations of miRNAs are determined to be significantly correlated [11]. Therefore, microRNAs have emerged as stable and sensitive diagnostic blood biomarkers for miscellaneous liver injuries [12]. Studies focusing on inflammatory liver damage in mouse models [13], steatohepatitis and hepatocarcinogenesis in human patients [14], as well as hepatobiliary diseases in dogs [12] have been revealed that serum miRNA profiling could be used as a valuable diagnostic, predictive, and prognostic tool to the degree of hepatic histologic abnormalities.

For that, this exploratory study aims to determine whether the expressions of selected miRNAs, based in previous studies, will be significantly measurable in the serum of dogs confirmed to have EHPSS or IHPSS. The study also aims to assess the potential of using selected serum miRNAs as a valuable addition in the current diagnostic and prognostic workup to truly reflect the nature of ongoing liver injuries in dogs with CPSS, which could be helpful for either clinical decision-making or monitoring the response after shunt occlusion in addition to the development of effective therapeutics using anti-miRNAs in the future.

## 2. Materials and Methods

### 2.1. Ethical Approval

The study was conducted after all owners signed a free and informed consent form for dogs to be included in the study and use of the data in publications. Animal care and procedures were carried out according to Directive 2010/63/EU of the European Parliament and of the Council of 22 September 2010 on the protection of animals used for scientific purposes. The study was conducted at Moscow State Academy of Veterinary Medicine and Biotechnology–MVA by K. I. Skryabin, Moscow, Russian Federation (protocol code 2-06-20-243).

### 2.2. Animals and Study Design

The present investigation was conducted on 96 dog cases of different pure breeds, body weight, and age at the Innovative Veterinary Center (IVC MVA), Moscow State Academy of Veterinary Medicine and Biotechnology, Moscow, Russia, in the period from September 2018 to October 2019. Suspected CPSS cases are tentatively diagnosed when dogs of predisposed breeds presented with a history of signs arousing clinical suspicion of CPSS, including neurologic signs half hour after feeding, intermittent vomition, as well as signs of renal calculi as stranguria with hematuria [15]. Eighty-four dogs were confirmed to have CPSS when all suspected cases underwent compendious CTA by using a 16-slice multidetector computed tomography scanner (Somatom Emotion^®^, Siemens, Germany) immediately after the ultrasonographic examination to characterize the morphology of shunts [16]. Based on our CTA findings (Figure 1), the investigated cases, as represented in Table 1, were assigned into five equal-sized groups (12 each) with EHPSS (splenocaval, splenophrenic, splenoazygos, right gastrocaval (RGC), and right gastrocaval with caudal loop (RGC–CL)) [17] and two equal-sized groups (12 each) with IHPSS (right and left divisional) [6]. For comparison, 12 healthy dogs of nearly matched breeds from cases that brought for periodic checkup were selected as a control group.

### 2.3. Sample Collection and Handling

All blood samples were gathered through the cephalic vein from each dog into a commercial plain test tube (2 mL Vacuette^®^ tube, Greiner Bio-One GmbH, Kremsmünster, Austria) that was left at room temperature for 20 min for maximum clot retraction and afterward the serum was separated by centrifugation at 2000 × *g* for 12 min at 20 °C in a CM-6M Sky Line centrifuge (ELMI^®^, Riga, Latvia). The separated serum was aliquoted into two Eppendorfs, the first one was directed for immediate biochemical analysis, whereas, the second one was preserved at −80°C in the absence of freeze–thaw cycles until miRNAs analysis. Additionally, intraoperative fresh wedge liver biopsies were gathered from each dog during the surgical attenuation of the shunt. Further, the healthy control dogs underwent a computed tomography (CT) guided core needle liver biopsies with the aid of a semi-automatic needle (spring cut^®^, Sterylab, Milan, Italy). The harvested liver biopsies were fixed in 10% neutral buffered formalin for further histopathological investigation.

### 2.4. Serum Biochemical Parameters Assays

The serum levels of total protein (TP), albumin (ALB), total bilirubin (T.BIL), alanine aminotransferase (ALT), aspartate aminotransferase (AST), alkaline phosphatase (ALP), gamma glutamyl transferase (GGT), amylase (AMYL), lipase (LIPA), total cholesterol (TC), triglycerides (TG), fasting ammonia (NH_3_), blood urea nitrogen (BUN), creatinine (CREA), and glucose (GLU) were estimated by using catalysts (Chem 17Clp^®^, NSAID 6 Clip^®^ and NH_3_ kits^®^, IDEXX Laboratories^®^, Westbrook, ME, USA) with the aid of a validated dry slide technology chemistry analyzer (Catalyst One^®^; IDEXX Laboratories, Westbrook, ME, USA), conferring to the manufacturer’s guidelines.

### 2.5. Histopathological Investigations

To this purpose, all liver biopsies were embedded in paraffin (Paraplast plus^®^, Brunswick, Mo., USA). For conventional histology, sections of 4-μm thickness were stained with hematoxylin and eosin (#RRSK26; Atom Scientific ^®^, Manchester, UK) [18]. Also, Perls’ Prussian blue (#PF 2617; Bio Diagnostic^®^, Giza, Egypt) was utilized to visualize hemosiderin pigments in hepatic tissue [19]. Moreover, Picrosirius red (#KT037; Diagnostic Bio Systems^®^, Hague, the Netherlands) was utilized to highlight connective tissue fibers (collagen types I and III) in investigated liver biopsies [20]. Histopathological evaluation applied by a veterinary pathologist according to the World Small Animal Veterinary Association standards [21] with the aid of a Mikmed-5 microscope^®^ (“LOMO”, Saint Petersburg, Russia).

### 2.6. RNA Isolation

Before RNA extraction, serum aliquots were scanned by spectrophotometry to test for hemolysis by measuring the free hemoglobin absorbance at 414 nm, and the samples with absorbance reading more than 0.18 were excluded from the study and replaced with another valid one [22]. Later, the total RNA, including miRNAs, was isolated from each sample using QIAzol lysis reagent as a part of the miRNeasy serum/plasma kit (Cat# 217184 Qiagen^®^, GmbH, Hilden, Germany) as per the manufacturer’s instructions. For measuring RNA isolation efficiency, 3.5 µL of synthetic *Caenorhabditis elegans* miRNA-39 (cel-miR-39, Cat# 219610, Qiagen^®^, GmbH, Hilden, Germany) was added as a spike-in control to each denatured serum sample at 1.6 ×10^8^ copies/µL working solution. After all, the concentration and purity of the isolated RNA samples were assessed by measuring their absorbance at 260 and 280 nm through using a spectrophotometer (NANOphotometer^®^ NP80, Implen, Germany), and the RNA ratios (A260:A280) more than and/or equal to 1.6 were included in the study [23].

### 2.7. Reverse Transcription and Quantification of miRNAs by real-Time Polymerase Chain Reaction (RT-PCR)

In brief, 5 µL of the isolated RNA from each sample was reverse transcribed using the miScript II RT kit (Cat #218160, Qiagen^®^, GmbH, Hilden, Germany) to prepare complementary DNA (cDNA) according to the manufacturer’s instructions. Later, the obtained cDNA (20 µl) immediately diluted in nuclease-free water (200 µl) to continue with RT–PCR. After that, to determine the relative expression of selected miRNAs and the exogenous cel-miR-39, RT–PCR was performed on the CFX-96^®^ real-time PCR detection system (Bio-Rad, USA) using 1 µL of the diluted cDNA and a miScript SYBR Green PCR kit (Cat#218073 Qiagen^®^, GmbH, Hilden, Germany) according to the miScript-PCR-System-Handbook. The PCR amplification reaction conditions were 1 cycle of initial denaturation at 95 °C for 15 min and 45 cycles of 3-step PCR, including 15 s of denaturation at 94 °C, annealing phase at 55 °C for 30 s and then an elongation phase for 30 s at 70 °C. All assays were applied in triplicate for each sample, and the average threshold cycle (C_T_) was estimated and used in subsequent analysis. Finally, the PCR-derived C_T_ values of each endogenous cfa-miRNA were normalized against cfa-miR-16 as an endogenous reference gene (due to their relatively stable expression) [24], and cel-miR-39 as an exogenous synthetic reference gene (spiked in during RNA isolation), ΔCt = CtmiRNA − 0.5 × (Ct *cel-miR-39* + Ct miR-16) [25] then all data were relatively expressed as a fold change compared to the controls by using the comparative C_T_ method (2^-ΔΔCt^ method) [26]. Specific forward primers to cfa-miRNAs of interest as well as endogenous reference gene were designed and synthesized by Evrogen Company (Moscow, Russia), as illustrated in Table 2, while the specific forward primer of cel-miR-39 and the universal reverse primer were already included in the commercially purchased kits (Qiagen^®^). Both the spike-in and the reference genes were not included in the statistical analysis but only utilized for data normalization.

### 2.8. Statistical Analysis

Data were statistically analyzed using SPSS version 20.0 (IBM Corp., NY, USA). In this study, the continuous variables that were assessed included age, body weight, serum biochemical parameters, serum miRNA expression levels, while the categorical variable included the breed, gender, and clinical signs. Firstly, the Shapiro–Wilk test was used to assess the normality distribution of the continuous variables and all of them were normally distributed (*p* > 0.05) except the cfa-miRNA expression levels (*p* < 0.05). Thereupon, the mean ± standard deviation (SD) was used to describe the normally distributed continuous variables and the significant differences between groups were determined by applying one-way ANOVA with post hoc testing and Bonferroni correction [27], whereas, the median and range were used to describe the miRNA expression levels and the significant differences between groups were determined by a Kruskal–Wallis test with post hoc testing and Bonferroni correction for *p*-value adjustment. Afterward, the diagnostic value of each differentially expressed miRNA to discriminate between CPSS groups and the control one was identified by calculating the area under the receiver operating (ROC) curve (AUC), and the cut-off values to obtain the optimum sensitivity and specificity percentage. For all analyses, *p*-values were two-sided and the significance level was set at *p* < 0.05.

## 3. Results

### 3.1. Characteristics of Dogs

Table 1 depicts that various morphologies of EHPSS were mostly noticed in small pedigree dogs such as Yorkshire terrier (most common breed), Russian toy terrier (common in the splenophrenic type), Miniature dachshund, Miniature pincher, Papillon, Shih Tzu, Pug, and Jack Russell terrier. Instead, the two morphologies of IHPSS were only diagnosed in large pedigree dogs such as Golden retriever and Labrador retriever. Notably, most of the dogs affected either with EHPSS or IHPSS were presented with signs of intermittent vomition, whereas, neurological signs after feeding as well as the sign of stranguria and hematuria were most commonly reported in dogs with splenocaval, RGC and RGC-CL shunts as well as the two morphologies of IHPSS. At the same time, dogs affected either with splenophrenic and splenoazygos shunts surprisingly diagnosed at a mean age significantly higher than all examined groups, while dogs affected with the two morphologies of IHPSS diagnosed at an age significantly lower than all investigated groups. Moreover, dogs affected with splenocaval, RGC and RGC–CL shunts presented at mean body weight significantly lower in comparison with the other examined groups.

### 3.2. Biochemical Results

Table 3 revealed that the serum activities of ALT, AST, ALP, and GGT were significantly elevated in dogs with splenocaval, RGC and RGC–CL shunts, and both types of IHPSS with respect to the splenophrenic, splenoazygos, and control groups. It is also observed that the serum levels of TP, ALB, globulin, BUN, TC, and TG were significantly declined. While ammonia concentration was significantly elevated in the dogs with splenocaval, RGC, RGC–CL, and the two morphologies of IHPSS in comparing with splenophrenic, splenoazygos, and control groups. Despite this, the serum value of A/G ratio, T.BIL, CREA, as well as the serum activities of LIPA and AMYL, were insignificantly varied within all investigated groups. As has been noted, all measured biochemical parameters were insignificantly varied in dogs either with splenophrenic or splenoazygos shunts when compared to the control group.

### 3.3. Histopathological Results

Liver biopsies from the control group confirmed the healthy status of the liver by showing apparent normal hepatocytes and normal histological architecture. Regardless of the location of the shunting vessel, the examined hepatic tissues from dogs with EHPSS or IHPSS, showed almost similar hepatocellular and vascular pathologies, albeit varying in frequency in each group. Microvesicular steatosis, accumulated hemosiderin in lipid granulomas, hepatocytes and/or sinusoidal Kupffer cells, together with the portal vein absence or hypoplasia were the most predominant lesion in all investigated groups, whereas lipid granuloma of different sizes, macrovesicular steatosis, and fibrosis more frequently expressed in liver biopsies from dogs with splenocaval, RGC, and RGC–CL shunts. Comparatively, arteriolar and biliary hyperplasia more frequently pronounced in dogs with two morphologies of IHPSS. Above all, previously mentioned hepatic lesions were minimally observed in dogs affected either with splenophrenic or splenoazygos shunts (Figure 2; Table 4).

### 3.4. Expression Profile of the Studied Cfa-Mirnas in the Investigated Groups Compared to the Control Group

Only cfa-miR-122 was significantly upregulated in all diseased groups compared to the control (Figure 3A), corresponding to a median fold change of 9.47, 9.41, and 9.21 in right divisional, splenocaval, and left divisional groups, respectively (all *p* < 0.001); 8.77 and 8.39 in RGC and RGC–CL groups, respectively (*p* < 0.01); 5.10 and 5.67 in splenophrenic and splenoazygos groups, respectively (*p* < 0.05). In a like manner, cfa-miR-34a and cfa-miR-21 were significantly expressed in the serum of dogs with splenocaval (median fold change = 10.75 and 5.69, respectively, *p* < 0.001), RGC (median fold change = 6.73 and 2.97, respectively, *p* < 0.01), RGC–CL (median fold change = 7.14 and 2.58, respectively, *p* < 0.01), right divisional (median fold change = 5.10 and 2.40, respectively, *p* < 0.05), and left divisional shunts (median fold change = 5.30 and 2.36, respectively, *p* < 0.05) compared to the control (Figure 3B,C). Uniquely, cfa-miR126 more significantly expressed (*p* < 0.001) in the serum of dogs within right divisional (median fold change = 11.81) and left divisional (median fold change = 10.87) groups, but less significantly expressed in RGC (median fold change = 9.40, *p* < 0.01), RGC–CL (median fold change = 5.48, *p* < 0.01) and splenocaval groups (median fold change= 4.61, *p* < 0.05) with respect to normal control (Figure 3D). At the same time, the levels of cfa-miR-34a, cfa-miR-21, and cfa-miR126 were insignificantly expressed (*p* > 0.05) in the serum of dogs either with splenophrenic or splenoazygos shunts compared with the control.

### 3.5. ROC Curve Analysis of Differentially Expressed Serum miRNAs

ROC curve analysis revealed that all examined miRNAs could be used as a novel diagnostic serum biomarkers to differentiate splenocaval shunts from the healthy controls with an AUC of 0.99 for cfa-miR-122 (95% CI: 0.97–1.0, *p* < 0.0001), 0.96 for cfa-miR-34a (95% CI: 0.86–1.0, *p* < 0.0001), 0.85 for cfa-miR-21 (95% CI: 0.68–1.0, *p* < 0.01), and 0.88 for cfa-miR-126 (95% CI: 0.75–1.0, *p* < 0.001), respectively (Figure 4A).The optimal sensitivity and specificity were (100% and 91.7% at cut-off 1.44), (91.7% and 91.7% at cut-off 1.23), (75% and 91.7% at cut-off 1.23), and (83.3% and 75% at cut-off 1.14) for serum cfa-miR-122, cfa-miR-34a, cfa-miR-21, and cfa-miR-126, respectively. Moreover, serum cfa-miR-122 with an AUC of 0.97 (95% CI: 0.92–1.0, *p* < 0.0001) and 0.96 (95% CI: 0.90–1.0, *p* < 0.0001) showed a potential role in differentiating splenophrenic and splenoazygos groups, respectively, from the control (Figure 4B,C), and the optimal sensitivity and specificity were (91.7% and 83.3% at cut-off 1.35), (91.7% and 83.3% at cut-off 1.32), respectively.

Furthermore, the differentially expressed serum miRNAs reflected a reasonable differentiation between RGC or RGC–CL groups and the control one by yielding AUC of 0.97, (95% CI: 0.89–1.0, *p* < 0.0001) for cfa-miR-122 in both groups and AUC of 0.91 (95% CI: 0.78–1.0, *p* < 0.001) and 0.89 (95% CI: 0.76–1.0, *p* < 0.001) for cfa-miR-34a, respectively; AUC of 0.87 (95% CI: 0.77–1.0, *p* < 0.01) and 0.85 (95% CI: 0.70–1.0, *p* < 0.01) for cfa-miR-21, respectively; AUC of 0.88 (95% CI: 0.73–1.0, *p* < 0.01) and 0.88 (95% CI: 0.75–1.0, *p* < 0.01) for cfa-miR-126, respectively (Figure 4D,E). The optimal sensitivity and specificity in RGC and RGC-CL were 91.7% and 100% at cut-off 1.60, 91.7% and 91.7% at cut-off 1.45 for cfa-miR-122; 83.3% and 66.7% at cut-off 1.15, 83.3% and 66.7% at cut-off 1.14 for cfa-miR-34a; 75% and 66.7% at cut-off 1.16, 83.3% and 66.7% at cut-off 1.07 for cfa-miR-21; and 75% and 75% at cut-off 1.14, 91.7% and 66.7% at cut-off 1.10 for cfa-miR-126, respectively.

In other hand, ROC analysis showed that cfa-miR-126 (AUC 1.00, 95% CI: 1.0–1.0, *p* < 0.0001) possessed higher diagnostic performance (100% sensitivity and 91.7% specificity at cut-off 1.28) than cfa-miR-122 (AUC 0.97, 95% CI: 0.89–1.0, *p* < 0.001, 91.7% sensitivity and specificity at cut-off 1.45), cfa-miR-34a (AUC 0.95, 95% CI: 0.87–1.0, *p* < 0.001, 91.7 % sensitivity and specificity at cut-off 1.23) and cfa-miR-21 (AUC 0.93, 95% CI: 0.83–1.0, *p* < 0.001, 91.7 % sensitivity and 67.3 specificity at cut-off 1.15) in discriminating the right or left divisional IHPSS group from the control group.

## 4. Discussion

Clinical characteristics of dogs included in our study are largely consistent with the data reported in previous literature [28]. Indeed, we noticed that congenital EHPSS is mostly diagnosed in small pedigree dogs, which may be reasoned to the aberrant lower hepatic expression of vascular cell adhesion molecule gene (VCAM1) that plays a potential role in the development of intrahepatic portal vascularization, whereas IHPSS was mostly observed among large breeds, which may be related to the hepatic over-expression of WEE1 gene that results in suppressing the post-natal closure of ductus venosus [29]. In this instance, Yorkshire terriers were the most predominant among breeds that are confirmed to have various morphologies of EHPSS except the splenophreic shunt, which is more frequently diagnosed in black Russian toy terrier and frequently observed in some cases in Moscow but not published in other studies up to the present time. Whereas IHPSS is most commonly detected in Golden and Labrador retrievers [30], we further attribute the dominance of these predisposed breeds among our cases as they are considered to be the most popular as companion dogs in Moscow.

In other words, cases with splenocaval, RGC, RGC–CL, and IHPSS mostly presented to our clinic with severe clinical signs (gastrointestinal, nervous, and urinary signs) and at a significantly earlier age in comparison with splenophreic and splenoazygos groups. That might belong to the complete diversion of portal vein engorged with ammonia (neurotoxin) and other toxic metabolites into the systemic circulation, particularly in those types of shunts, as confirmed by our CTA and previous investigations [17,28,31], hence, increasing the severity of the disease to be noticed at an early age. On the contrary, dogs with splenophrenic or splenoazygos shunts were mostly identified by chance, as in spite of being a congenital disease, they presented at a significantly older age with mild clinical signs, together with significantly higher body weight than other EHPSS types, which could be linked to the low severity of those types of shunts [17,28]. After all, in common with previous studies, we did not observe a sex linkage in our study due to the occurrence of all CPSS forms in both genders [9].

Correspondingly, the results of the present study demonstrate that the serum activity of hepatic enzymes and the serum level of ammonia are significantly elevated, while the serum level of TP, ALB, Globulin, BUN, TC, and TG are significantly declined, in dogs with splenocaval, RGC, RGC–CL, and both types of IHPSS in comparison with splenophrenic, splenoazygos, and control groups. This result was concordant with the previous retrospective study [17,32,33], which clarified that the severity of liver dysfunction and serum biochemical abnormalities in dogs with CPSS usually parallel to the amount of blood diverted from the portal circulation to the systemic circulation without liver perfusion, inducing hypoxic cellular damage and also enzyme leakage as a consequence of reducing the functional capacity of hepatocytes to perform their metabolic function.

In addition, our CTA revealed that the portal circulation is completely diverted into the CVC without hepatic perfusion in splenocaval, RGC RGC–CL, as well as right and left divisional groups. Henceforth, the severity of biochemical and hepatic histopathological changes in the present study was more significantly prevalent in those types of shunts in comparing with splenophrenic or spenozygos shunts which were insignificantly differed with the control since those types of shunts are easy to be compressed during each respiratory cycle and gastric distension after eating. Hence the liver takes the chance to be partially perfused with the stagnant portal blood, which could explain the cause behind subtle serum biochemical and histopathological abnormalities in these groups [28]. Also, the presented data clarified that the azygos vein appeared torturous in shape as it has less diameter and capacity than that of the CVC, which likely creates resistance to blood flow from the azygos vein into CVC, thereby facilitating the liver perfusion by taking advantage of this blood flow resistance in the shunting vessel. Of note, the insignificant variation of serum T.BIL, CREA value, as well as the serum LIPA and AMYL activities in the diseased groups compared to control, suggest the absence of concurrent jaundice, renal, and pancreatic damage, respectively [32].

Our study further confirmed the high sensitivity of the fasting serum ammonia concentration as a noninvasive screening biomarker to detect the presence of CPSS. It was outside the normal control range in all dogs with CPSS except dogs in the splenophrenic and splenoazygos groups as they have the advantage of reducing the amount of blood-engorged with ammonia to be diverted outside the liver into the systemic circulation [28]. However, in spite of the greater diagnostic benefit of biochemical markers and ammonia concentration, they could not always truly specify portosystemic shunts from other hepatic pathologies [34] and may remain normal or minimally altered in dogs with splenophrenic or splenoazygos shunts, and may also fail to reflect the severity of hepatic injury [35] and the ability of hepatic tissue to accommodate a sudden portal blood flow within it after shunt attenuation [5,9]. Henceforth, the development of accurate, reliable, and noninvasive liver-specific tools such as the molecular markers to accurately predict the pathological nature of ongoing liver injuries and monitor cases over time has long been desired.

So far, circulating hepatocyte derived miRNAs may represent attractive novel reliable diagnostic biomarkers for monitoring liver damage. They usually exhibit the great potential to address some of the significant limitations described for serum biochemical markers of liver injuries [11]. Also, they can even outperform serum ALT values as sensitive, specific, and predictive hepatic biomarkers in dogs [12,36,37] and human models [14].

The shunting vessel usually reduces the amount of oxygen and gastrointestinal-derived factors (such as insulin) to be delivered to hepatic tissue in dogs with CPSS which differ in intensity according to shunt location [38], hence producing intrahepatic portal vein absence or hypoplasia, hepatic hypoxic injury and a defect of fatty acid metabolism, resulting in microvesicular and macrovesicular steatosis [9] beside hemosiderin accumulation in vacuolated hepatocytes that are directly phagocytized by Kupffer cells [39] and subsequently aggregated in forms of lipogranulomas or pigment granulomas when hemosiderin (brown pigment) remain in phagocytized vacuolated hepatocytes [40]. Lipogranulomas (a remnant of degenerated lipid-filled hepatocytes) are observed to be the most common feature in dogs affected with splenocaval, RGC, RGC–CL, and minimally observed in biopsy samples from both types of IHPSS, as the lipogranuloma usually provides circumstantial evidence not only for the degree of hepatic tissue hypoperfusion due to portal vein absence or hypoplasia but also for the disease persistence over a long period as it is correlated with the increasing age of the dog [41].

Nowadays, hepatocyte-derived miRNA-122 represents 70% of all miRNAs in the liver [42], and it plays a precise role in regulating hepatocyte development and differentiation, and is also involved in homeostasis between fatty acids and cholesterol biosynthesis in hepatocytes; hence, circulating miR-122 could be an early serum biomarker for the degree of hepatocytes oxidative stress, necroinflammatory activity, and simple hepatic steatosis [43]. Moreover, miR-122 is a valuable specific and sensitive serum biomarker for the diagnosis of various canine hepatic pathologies [36,44]. In this regard, as one of the most abundant miRNAs in the liver, cfa-miR-122, was the only miRNA to be significantly upregulated in all CPSS groups compared to the control with a median fold change of *p*-value < 0.001 in right divisional, splenocaval, and left divisional groups; *p*-value < 0.01 in RGC and RGC–CL groups and *p* < 0.05 in splenophrenic and splenoazygos groups with high sensitivity and specificity according to AUC yield in each group. In particular, miR-122 upregulation was significantly associated with the presence of microvesicular steatosis that determined to be the most common liver injury in all investigated groups, including splenoazygos and splenophrenic groups. Moreover, we attributed the significant upregulation of serum cfa-miR-122 in splenoazygos or splenophrenic group with respect to the control group to the fact that cfa-miR-122 usually reveals high diagnostic performance in early diagnosis of mild hepatic steatosis in those groups than all other routinely used biochemical parameters, including ALT [45].

Hepatocyte-derived miR-34a has emerged as a potent regulator of steatohepatitis progression via repressing SIRT1 gene that acts as a regulator of energy metabolism within hepatocytes by controlling of AMP kinase activity, hence repressing of SIRT1 gene may result in decreasing hepatocyte metabolism, increasing fat oxidation, progression of hepatic steatosis, and finally cell death [42,46]. Upon this, cfa-miR-34a was significantly expressed in serum of dogs with splenocaval (*p* < 0.001), RGC (*p* < 0.01), RGC–CL (*p* < 0.01), right divisional (*p* < 0.05), and left divisional shunts (*p* < 0.05) compared to control. In addition, reflecting considerable diagnostic sensitivity and specificity according to ROC curve analysis, but already less than cfa-miR-122 and also their expression, cannot discriminate against the splenophrenic or splenoazygos groups from the healthy control. These findings were linked to the frequent prevalence of vacuolated hepatocytes with macrovesicular steatosis, as well as lipid or pigment granuloma, which were particularly frequent in splenocaval, RGC, RGC–CL, right divisional and left divisional shunts and not observed in dogs with splenophrenic or splenoazygos shunts.

Occasionally, hepatic hypoxia in a dog with CPSS could enhance the accumulation of the adenosine (vasodilator) within the space of mall via autoregulation [47,48], hence promote hepatic arteriolar proliferation that is usually associated with ductular hyperplasia in attempt to meet the oxygen demands of hepatocytes and gastrointestinal tract–derived growth factors, insulin, and nutrients [38,49]. By that time and with increasing dog age, increased arteriolar proliferation cannot completely compensate reduced portal blood supply in the dog with CPSS, leading to chronic liver injuries and sustained hepatic stellate cell (HSCs; Ito cells) activation, which are important producers of perisinusoidal extracellular matrix (ECM) accumulation during fibrogenesis, hence inducing excessive collagen deposition (collagen types I and III) in the portal and parenchymal area [47,50,51].

Recent studies have defined hepatocyte-derived miR-21 as one of the main driving forces for the progression of hepatic damage into fibrosis as it indirectly activates HSC for ECM synthesis via activation of TGF-β1/Smads, ERK, PTEN/Akt, and NF-κB signaling pathways [52,53]. In addition, serum miR-21 has been reported to be prevalently upregulated during pathogenesis and progression of liver fibrosis in human and mouse models [54]. Likewise, cfa-miR-21 was markedly over-expressed, corresponding with the portal and parenchymal fibrosis in each group, as it was significantly elevated in the serum of dogs with splenocaval (*p* < 0.001), RGC (*p*<0.01), RGC–CL (*p* < 0.01), right divisional (*p* < 0.05) and left divisional shunts (*p* < 0.05), with high sensitivity and specificity in differentiating diseased groups from the control, according to AUC yield.

Interestingly, endothelial specific microRNA (miR-126) play an important role in ischemic reperfusion injury by promoting neovascularization (arteriogenesis) in response to vessel injury and/or tissue hypoxia for improving cell survival [55,56]. Recent insights underline the precise role of miR-126 in reducing apoptosis and fibrosis by promoting angiogenesis of endothelial cells via the PI3K/eNOS/NO pathway, suppressing oxidative stress and stabilizing mitochondrial membrane potential in vivo [57]. Furthermore, miR-126 could act as a diagnostic and screening biomarker for liver neoplastic disease as it has been overexpressed in the serum of dogs with hepatocellular carcinoma [58] due to its critical roles on tumor angiogenesis [59]. In the same way, we observed that cfa-miR126 was significantly expressed in serum with fold change and *p*-value related to the frequency of the altered hepatic blood supply in the form of arteriolar proliferation and its associated ductular proliferation in our investigated groups, where it more significantly upregulated (*p* < 0.001) in the serum of dogs with both types of IHPSS but less significantly expressed in dogs with RGC (*p* < 0.01), RGC-CL (*p* < 0.01), and splenocaval shunts (*p* < 0.05), with respect to normal control. Additionally, according to ROC analysis, cfa-miR126 revealed high diagnostic performance with different AUC yield, sensitivity, and specificity, corresponding to the frequency of the altered hepatic blood supply in each diseased group.

As an illustration, one previous study mentioned that there was not sufficient hepatocellular damage in the dog with CPSS to overexpress miR-122, miR-21, and miR-126 in the serum. These results were based upon only six cases of CPSS, which are not ideal representative numbers [36], whereas our study included a representative number of dogs with various shunt morphologies accompanied by histopathological abnormalities. In other words, we could assume that these six cases may mostly be affected by the less severe shunt types (splenoazygos or splenophrenic) [28] that usually show minimal hepatic damage and insignificant expression of serum miRNAs, except miR-122, as we determined in our investigation.

## 5. Conclusions

Despite the greater diagnostic benefit of serum ammonia concentration to predict the presence of portosystemic shunt and serum biochemical parameters to reveal hepatic damage, they could not truly specify the nature of ongoing hepatic pathologies and remained normal in dogs with splenophrenic or splenoazygos shunts compared to the control. Therefore, we provide new evidence that serum miRNAs (cfa-miR-122, cfa-miR-34a, cfa-miR-21, and cfa-miR-126) could be potentially expressed in different CPSS groups and could serve as novel non-invasive biomarkers to mirror the histopathological and molecular events occurring in the liver in each shunt type, with high sensitivity and specificity. That could be helpful in the future either for the development of effective therapeutics using anti miRNAs or for clinical decision-making of surgical intervention or for monitoring the response after surgical occlusion of shunt without the current need for liver biopsies, which have potential complications.

## Figures and Tables

**Figure 1 vetsci-07-00035-f001:**
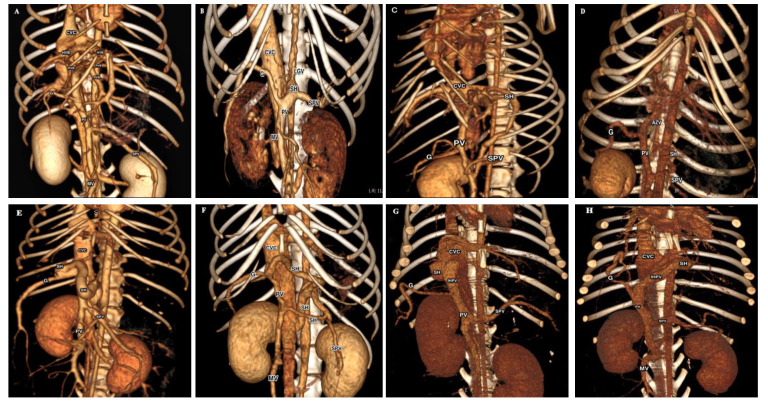
Three-dimensional volume-rendered images (ventral view) of normal portal venous system in control group (**A**); five common EHPSS morphologies (splenocaval (**B**); splenophrenic (**C**); splenoazygos (**D**); RGC (**E**); RGC-CL (**F**)) and two common IHPSS morphologies (right divisional (**G**) and left divisional (**H**)). *CVC*, caudal vena cava; *HVB*, hepatic vein branch; *PVB*, portal vein branch; *G*, right gastroduodenal vein; *SH*, shunt; *SPV*, splenic vein; *PV*, portal vein; *MV*, mesenteric vein; *LGV*, left gastric vein; *AZV*, azygos vein; *IHPV*, intrahepatic portal vein.

**Figure 2 vetsci-07-00035-f002:**
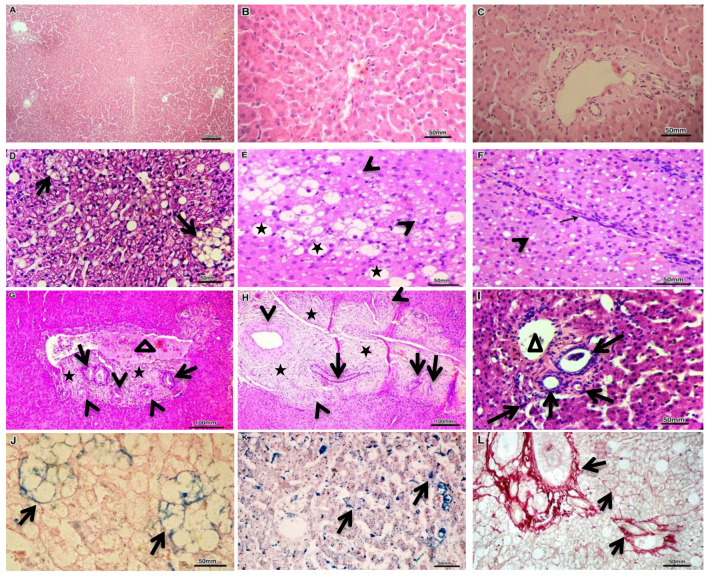
Representative histopathology from dogs affected either with EHPSS or IHPSS, as well as the healthy control ones. H&E stained normal liver sections with normal liver lobules and central vein in its center from which radiate one cell thick liver cell plates towards the portal field, which include hepatic artery, bile duct, and portal vein **(A–C)**. H&E stained liver sections showed various forms of fatty degeneration, including lipid granuloma (**D**, **arrows**), macrovesicular steatosis (**E**, **stars**), and microvesicular steatosis (**E**,**F**, **arrowhead**) as well as thick intralobular fibrous septa with lymphocytic infiltration (**F**, **arrow**). H&E stained liver sections declare the portal area expansion by fibrous tissue deposition (**G**,**H**, **stars**), biliary hyperplasia (**G**–**I**, **arrows**) and arteriolar proliferation (**G**,**H**, **arrowhead**) together with partially occluded portal vein (**G**, **triangle**), absence of portal vein (**H**) and hypoplastic (**small caliber**) portal vein (**I**, **triangle**). Perls’ Prussian blue-stained iron salts (hemosiderin’s) within lipid granuloma (**J**, **arrows**) as well as hepatocytes and sinusoidal Kupffer cells (**K**, **arrows**). Picrosirius red-stained fibers of connective tissue (collagen types I and III) in portal and parenchymal areas (**L**, **arrows**).

**Figure 3 vetsci-07-00035-f003:**
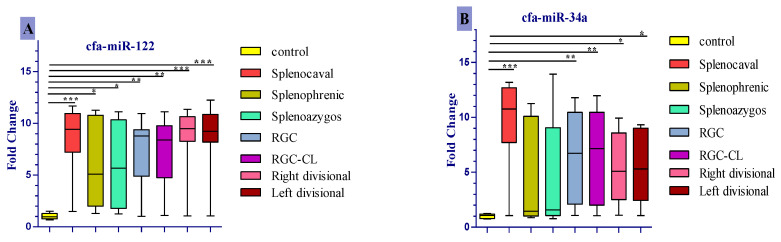
Comparison of expression level of cfa-miR- 122 (**A**), cfa-miR- 34a (**B**), cfa-miR-21 (**C**), cfa-miR-126 (**D**) in serum of dogs affected either with EHPSS or IHPSS as well as the healthy control ones. Significant differences between CPSS groups and the normal control group are marked with stars (* *p* < 0.05, ** *p* < 0.01, *** *p* < 0.001).

**Figure 4 vetsci-07-00035-f004:**
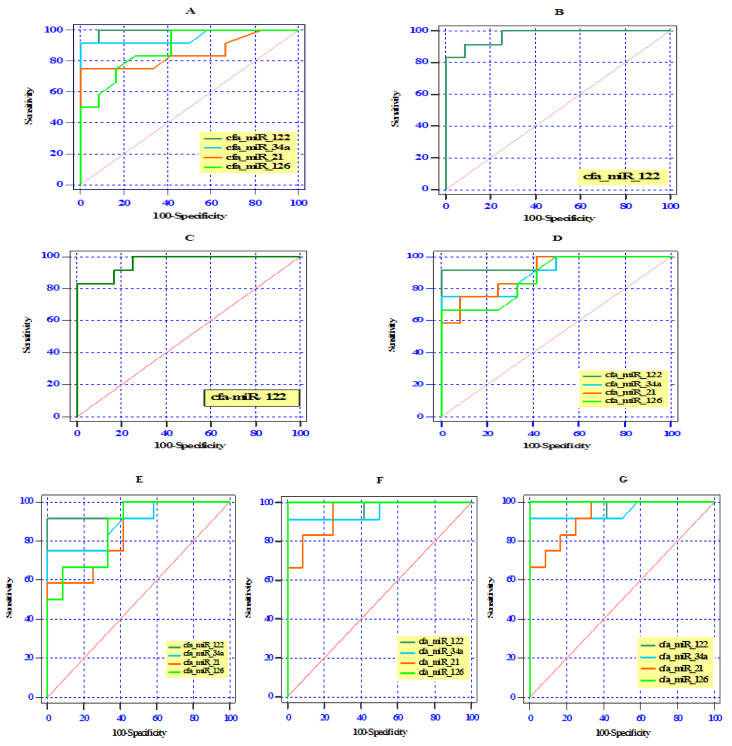
Receiver operating (ROC) curve analysis of the differentially expressed cfa-miRNAs among the studied groups (splenocaval (**A**), splenophrenic (**B**), splenoazygos (**C**), RGC (**D**), RGC–CL (**E**), right divisional (**F**), and left divisional (**G**) and the control group.

**Table 1 vetsci-07-00035-t001:** Dog characteristics included in the study.

Groups.		EHPSS					IHPSS	
Variables	Control (n = 12)	Splenocaval (n = 12)	Splenophrenic (n = 12)	Splenoazygos (n = 12)	RGC (n = 12)	RGC-CL (n = 12)	Right Divisional (n = 12)	Left Divisional (n = 12)
**Breed (NO.)**								
Yorkshire terrier	(3)	(4)	(2)	(4)	(5)	(4)	(0)	(0)
Miniature dachshund	(1)	(3)	(2)	(1)	(1)	(2)	(0)	(0)
Miniature pincher	(1)	(1)	(0)	(2)	(0)	(0)	(0)	(0)
Russian toy terrier	(2)	(0)	(5)	(0)	(0)	(1)	(0)	(0)
Papillon	(1)	(0)	(1)	(1)	(2)	(2)	(0)	(0)
Shih Tzu	(0)	(1)	(1)	(2)	(2)	(2)	(0)	(0)
Pug	(0)	(1)	(0)	(1)	(1)	(0)	(0)	(0)
Jack Russell terrier	(1)	(2)	(1)	(1)	(1)	(1)	(0)	(0)
Golden Retriever	(2)	(0)	(0)	(0)	(0)	(0)	(8)	(9)
Labrador retriever	(1)	(0)	(0)	(0)	(0)	(0)	(4)	(3)
**Clinical symptoms (NO.)**								
Neurologic signs after feeding	(0)	(11)	(2)	(1)	(9)	(9)	(11)	(12)
Intermittent vomition	(0)	(12)	(9)	(9)	(10)	(8)	(12)	(12)
Stranguria and hematuria	(0)	(9)	(1)	(2)	(8)	(10)	(9)	(11)
**Gender (NO.)**	F(3), FS(1), M(5), MN(3)	F(2), FS(1), M(8), MN(1)	F(1), FS(2), M(7), MN(2)	F(2), FS(0), M(7), MN(3)	F(4), FS(0), M(7), MN(1)	F(4), FS(1), M(6), MN(1)	F(1), FS(1), M(9), MN(1)	F(2), FS(0), M(9), MN(1)
**Age (months), Mean ± SD**	28.95 ± 10.72 ^b^	22.85 ± 6.47 ^b^	39.83 ± 9.31 ^a^	40.58 ± 13.05 ^a^	22.58 ± 7.34 ^b^	20.33 ± 6.30 ^b^	6.58 ± 1.78 ^c^	6.67 ± 1.92 ^c^
**Body weight (kg), Mean ± SD**	5.12 ± 0.93 ^a^	3.69 ± 0.67 ^b^	4.98 ± 0.80 ^a^	5.01 ± 0.98 ^a^	3.78 ± 1.01 ^b^	3.70 ± 0.91 ^b^	5.12 ± 1.07 ^a^	5.29 ± 1.02 ^a^

^a,b,c^ Variables with different superscripts in the same row are significantly different at *p* < 0.05. EHPSS, extrahepatic portosystemic shunts; IHPSS, intrahepatic portosystemic shunts; RGC, right gastrocaval; RGC–CL, right gastrocaval with caudal loop; F, female; FS, female spayed; M, male; MN, male neutered.

**Table 2 vetsci-07-00035-t002:** Mature sequences and specific forward primers for endogenous reference and target miRNAs.

Gene Name	Accession Number	Mature Sequence 5′–3′	Primer Sequence 5′–3′
cfa-miR-122-5p	MIMAT0006619	UGGAGUGUGACAAUGGUGUUUG	CAAACACCATTGTCACACTCCA
cfa-miR-34a-5p	MIMAT0006690	UGGCAGUGUCUUAGCUGGUUGU	ACAACCAGCTAAGACACTGCCA
cfa-miR-21-5p	MIMAT0006741	UAGCUUAUCAGACUGAUGUUGA	TCAACATCAGTCTGATAAGCTA
cfa-miR-126-5p	MIMAT0006730	CAUUAUUACUUUUGGUACGCG	CGCGTACCAAAAGTAATAATG
cfa-miR-16-5p	MIMAT0006648	UAGCAGCACGUAAAUAUUGGCG	CGCCAATATTTACGTGCTGCTA

cfa-miR, canine familiaris microRNA.

**Table 3 vetsci-07-00035-t003:** Biochemical panel in dogs affected either with extrahepatic or intrahepatic portosystemic shunts (EHPSS or IHPSS) as well as the healthy control ones.

Group	Control	EHPSS					IHPSS	
Variable		Splenocaval	Splenophrenic	Splenoazygos	RGC	RGC-CL	Right Divisional	Left Divisional
**ALT (U/L)**	44.59 ± 8.35 ^b^	134.16 ± 21.57 ^a^	68.98 ± 12.80 ^b^	66.28 ± 14.41 ^b^	120.16 ± 18.14 ^a^	123.52 ± 23.33 ^a^	143.64 ± 31.94 ^a^	139.32 ± 20.73 ^a^
**AST (U/L)**	34.07 ± 4.76 ^b^	88.37 ±8.95 ^a^	46.28 ± 7.44 ^b^	45.24 ± 6.53 ^b^	77.22 ± 14.60 ^a^	81.91 ± 17.12 ^a^	92.05 ± 8.71 ^a^	90.03 ± 7.55 ^a^
**ALP (U/L)**	76.15 ± 11.31 ^b^	292.1 ± 55.45 ^a^	100.65 ± 19.83 ^b^	102.25 ±18.22 ^b^	271.21 ± 30.15 ^a^	275.13 ± 26.24 ^a^	307.93 ± 31.22 ^a^	304.59 ± 30.17 ^a^
**GGT (U/L)**	7.20 ± 1.70 ^b^	12.07 ± 1.57 ^a^	8.06 ± 1.36 ^b^	8.24 ± 1.18 ^b^	10.54 ± 1.88 ^a^	10.73 ± 1.49 ^a^	12.28 ± 1.58 ^a^	12.16 ± 1.35 ^a^
**TP(g/dL)**	6.47 ± 0.72 ^a^	3.73 ± 0.82 ^b^	5.70 ± 0.59 ^a^	5.74 ± 0.70 ^a^	3.94 ± 0.46 ^b^	3.85 ± 0.43 ^b^	3.32 ±0.45 ^b^	3.20 ± 0.68 ^b^
**ALB (g/dL)**	3.00 ± 0.58 ^a^	1.98 ± 0.40 ^b^	2.78 ± 0.43 ^a^	2.72 ± 0.70 ^a^	2.09 ± 0.37 ^b^	2.03 ± 0.29 ^b^	1.64 ± 0.38 ^b^	1.60 ± 0.45 ^b^
**Globulin (g/dL)**	3.46 ± 0.42 ^a^	1.75 ± 0.63 ^b^	2.92 ± 0.69 ^a^	2.99 ± 0.36 ^a^	1.84 ± 0.33 ^b^	1.82 ± 0.40 ^b^	1.69 ± 0.56 ^b^	1.59 ± 0.47 ^b^
**A/G ratio**	0.88 ± 0.21 ^a^	1.26 ± 0.46 ^a^	1.01 ± 0.32 ^a^	0.94 ± 0.28 ^a^	1.18 ± 0.32 ^a^	1.17 ± 0.36 ^a^	1.14 ± 0.69 ^a^	1.10 ± 0.53 ^a^
**T.BIL (μmol/L)**	4.64 ± 0.98 ^a^	5.27 ± 0.85 ^a^	4.82 ±0.81 ^a^	4.74 ± 0.72 ^a^	5.24 ± 0.75 ^a^	5.31 ±0.67 ^a^	5.32 ± 0.88 ^a^	5.36 ± 0.83 ^a^
**BUN (μmol/L)**	5.78 ± 0.77 ^a^	2.49 ± 0.40 ^b^	5.40 ± 1.68 ^a^	5.51 ± 1.57 ^a^	2.98 ± 1.05 ^b^	2.88 ± 1.43 ^b^	2.25 ± 0.59 ^b^	2.29 ± 0.37 ^b^
**Ammonia (μmol/L)**	25.31 ± 6.23 ^b^	230.80 ± 67.83 ^a^	76.29 ± 14.44 ^b^	70.32 ± 17.25 ^b^	205.70 ± 29.28 ^a^	211.45 ± 29.85 ^a^	253.48 ± 56.67 ^a^	260.26 ± 63.04 ^a^
**TC (μmol/L)**	5.02 ± 1.02 ^a^	2.51 ± 0.50 ^b^	4.02 ± 1.07 ^a^	4.15 ± 1.17 ^a^	2.73 ± 0.81 ^b^	2.81 ± 0.77 ^b^	2.44 ± 0.46 ^b^	2.26 ± 0.51 ^b^
**TG (μmol/L)**	0.75 ± 0.23 ^a^	0.33 ± 0.16 ^b^	0.66 ± 0.22 ^a^	0.68 ± 0.20 ^a^	0.37 ± 0.13 ^b^	0.38 ± 0.11 ^b^	0.30 ± 0.14 ^b^	0.28 ± 0.09 ^b^
**GLU (mg/dl)**	4.20 ±0.93 ^a^	2.35 ± 0.77 ^b^	3.68 ± 0.49 ^a^	3.57 ± 0.56 ^a^	2.41 ± 0.61 ^b^	2.46 ± 0.55 ^b^	2.24 ± 0.66 ^b^	2.34 ± 0.57 ^b^
**CREA (μmol/L)**	49.33 ± 5.93 ^a^	48.67 ± 5.09 ^a^	46.53 ± 4.24 ^a^	47.68 ± 5.23 ^a^	48.88 ± 2.38 ^a^	46.43 ± 6.13 ^a^	50.34 ± 5.56 ^a^	48.54 ± 5.66 ^a^
**LIPA (U/L)**	131.75 ± 31.84 ^a^	128.25 ± 24.61 ^a^	129.67 ± 26.02 ^a^	130.42 ±25.66 ^a^	133.00 ± 31.87 ^a^	131.42 ± 37.64 ^a^	125.83 ± 34.56 ^a^	129.33 ± 27.49 ^a^
**AMYL (U/L)**	174.99 ± 59.44 ^a^	169.27 ± 97.28 ^a^	181.44 ± 54.76 ^a^	177.50 ± 93.67 ^a^	167.56 ± 60.19 ^a^	173.32 ± 60.51 ^a^	183.62 ± 78.73 ^a^	179.96 ± 68.96 ^a^

^a, b^ Variables with different superscript in the same row are significantly different at *p* < 0.05. EHPSS, extrahepatic portosystemic shunts; IHPSS, intrahepatic portosystemic shunts; RGC, Right gastrocaval; RGC–CL, Right gastrocaval with caudal loop; ALT, alanine aminotransferase; AST, aspartate aminotransferase; ALP, alkaline phosphatase; GGT, gamma glutamyl transferase; TP, total protein; ALB, albumin; A/G ratio, albumin/ globulin ratio; T.BIL, total bilirubin, BUN, blood urea nitrogen; TC, total cholesterol; TG, triglycerides; GLU; glucose; CREA, creatinine; LIPA, lipase; AMYL, amylase.

**Table 4 vetsci-07-00035-t004:** Hepatocellular and vascular pathologies in the liver biopsies from dogs affected either with EHPSS or IHPSS, as well as the healthy control ones.

Group	Control	EHPSS					IHPSS	
Histopathologic Feature (s)		Splenocaval	Splenophrenic	Splenoazygos	RGC	RGC–CL	Right Divisional	Left Divisional
Microvesicular steatosis	0 (0%)	12 (100%)	8 (66.67%)	9 (75%)	10 (83.33%)	9 (75%)	12 (100%)	12 (100%)
Macrovesicular steatosis	0 (0%)	10 (83.33%)	0 (0%)	1 (8.33%)	9 (75%)	8 (66.67%)	4 (33.33%)	3 (25%)
Lipid granuloma	0 (0%)	9 (75%)	0 (0%)	0 (0%)	7 (66.67%)	8 (66.67%)	2 (16.67%)	1 (8.33%)
Hemosiderin within lipid granuloma or Kupffer cells	0 (0%)	11 (91.67%)	1 (8.33%)	2 (16.67%)	10 (83.33%)	10 (83.33%)	11 (91.67%)	12 (100%)
Portal veins absence or hypoplasia	0 (0%)	12 (100%)	3 (25%)	3 (16.67%)	12 (100%)	12 (100%)	12 (100%)	12 (100%)
Arteriolar hyperplasia	0 (0%)	7 (58.33%)	2 (16.67%)	3 (25%)	9 (75%)	8 (66.67%)	12 (100%)	12 (100%)
Biliary hyperplasia	0 (0%)	7 (58.33%)	2 (16.67%)	3 (25%)	9 (75%)	8 (66.67%)	12 (100%)	12 (100%)
Fibrosis (portal, periportal and parenchymal)	0 (0%)	7 (58.33%)	2 (16.67%)	2 (16.67%)	6 (50%)	7 (58.33%)	3 (25%)	4 (33.33%)

EHPSS, extrahepatic portosystemic shunts; IHPSS, intrahepatic portosystemic shunts; RGC, right gastrocaval; RGC–CL, right gastrocaval with caudal loop.

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
