# Peer review of "Clinical Characteristics, Serum Biochemical Changes, and Expression Profile of Serum Cfa-miRNAs in Dogs Confirmed to Have Congenital Portosystemic Shunts Accompanied by Liver Pathologies"

_vetsci, 2020, doi:10.3390/vetsci7020035_

Round 1

Reviewer 1 Report

This manuscript deals with an interesting topic of searching novel biomarkers to diagnose congenital portosystemic shunts in dogs. The manuscript is well written. But, some concern needs to be addressed to be fit for publication as follows:
1. There is a problem in using the abbreviations throughout the manuscript. The abbreviation must be introduced upon the first mentioning of the full term followed by its abbreviation in parentheses: From then on, the abbreviation must be used exclusively and throughout. For example line 35: CPSS should be replaced with congenital portosystemic shunts (CPSS).
2. The characteristic of dogs should be transferred to the method section as the information of the screened animals should be presented first as this is not a result. Hence, table 2 should b table 1 and rearrange the tables.
3. Lin 92: the authors mentioned that control groups of matched breeds were used. However, according to the data in table 2, Shih Tzu and Pug breeds have no control dogs. Please, clarify.
4. Line 102: all samples. What samples? The authors should clarify that these are blood samples. Also, the authors should clarify how the blood samples were collected.
5. In the statistical analysis: as appeared in the presentation of the results, the authors have not taken the age, sex or breed as factors in the statistical analysis. There is some conflict with what mentioned in the statistical analysis. Also, in the discussion, the authors discussed the differences in the prevalence of breed, sex, and age. On what basis the authors have assessed the prevalence while these factors have not been taken into consideration in the statistical analysis??
6. In figure 2: all lesions demonstrated in the legend should be denoted by arrows or stars in the specified figures.
7. In all tables, all abbreviations used should be illustrated in the footnotes.

Reviewer 2 Report

The selection of new biomarkers is important for the rapid diagnosis of canine disease. This study provides detailed data to search for markers of hepatic injury associated with extrahepatic and intrahepatic portosystemic shunt (EHPSS and IHPSS).
1. Why were these 5 cfa-mirnas selected by the authors for detection and correlation analysis of liver injury associated with extrahepatic and intrahepatic portosystemic shunt (EHPSS and IHPSS)?
2. The evidence for evaluating liver injury by HE staining alone is rather thin, suggesting that the results of immunohistochemical tests should be increased.

Reviewer 3 Report

This is a well conducted study. Background and rationale are well described; the methodological approach is compatible with the aim of the study. Material and methods are well designed and results clearly  reported and deeply discussed

I have only minor comments.

In the discussion, there are some sentences difficult to understand; perhaps they need to be rephrased:

from line 377 tom 380: “ It is worthy to…..

from line 424 to line 438: the last paragraph of the discussion is not clear to me; perhaps references 26 and 38 are not correctly positioned since they do not help me to understand the meaning of the paragraph.

Minors

Tab 1 should be reformatted (miRNA mature and primer sequences for each miRNA should be written in the same row)

Tab 2 should be reformatted (the rows of AGE and of GENDER)

Line 55: for the first time the word microRNA is used. In parenthesis should be reported the term miRNA that is utilized in the remaining part of the manuscript.

Line 102: please specify the type of samples, blood samples

For qRT/PCR, how much RNA was reverse transcribed? And, how much cDNA for each sample was amplified by PCR?

Round 2

Reviewer 1 Report

The authors adequately responded to all comments and performed all the required modifications as directed.

Reviewer 2 Report

The author has answered all the questions and this manuscript is available for publication.